# A Distributed Anti-Jamming Algorithm Based on Actor–Critic Countering Intelligent Malicious Jamming for WSN

**DOI:** 10.3390/s22218159

**Published:** 2022-10-25

**Authors:** Yuheng Chen, Yingtao Niu, Changxing Chen, Quan Zhou, Peng Xiang

**Affiliations:** 1Fundamentals Department, Air Force Engineering University of People’s Liberation Army, Xi’an 710051, China; 2The Sixty-Third Research Institute, National University of Defense Technology, Nanjing 210007, China; 3College of Communications Engineering, Army Engineering University of People’s Liberation Army, Nanjing 210007, China

**Keywords:** actor–critic, distributed multi-agent, anti-jamming communication, Multi-Agent Markov Decision Process

## Abstract

In this paper, in order to solve the problem of wireless sensor networks’ reliable transmission in intelligent malicious jamming, we propose a Distributed Anti-Jamming Algorithm (DAJA) based on an actor–critic algorithm for a multi-agent system. The Multi-Agent Markov Decision Process (MAMPD) is introduced to model the progress of wireless sensor networks’ anti-jamming communication, and the multi-agent system learns the intelligent jamming from the external environment by using an actor–critic algorithm. On the basis of coping with the influence of external and internal factors effectively, each sensor in networks selects the appropriate channels for transmission and finally realizes the optimal transmission of the system overall in a unit time period. In the environment of probabilistic intelligent jamming with tracking properties set in this paper, the simulation shows that the algorithm proposed can outperform the algorithm based on joint Q-learning and the conventional scheme based on orthogonal frequency hopping in terms of transmission. In addition, the proposed algorithm completes two updates of strategy evaluation and action selection in one iteration, which means that the system has higher efficiency of action selection and better adaptability to the environment through the interaction with the external environment, resulting in the better performance of transmission and convergence.

## 1. Introduction

A wireless sensor network (WSN) is a loosely coupled multi-agent network composed of a large number of sensor nodes with multi-functions. It is widely used in many fields such as forming the foundation of the Internet of Things (IoT) [1,2]. However, due to the openness of WSNs and the complex electromagnetic environment a WSN is always working in, it is vulnerable to all kinds of interference and jamming, especially external malicious jamming from enemy entities. The malicious jamming can seriously affect the reliability and effectiveness of the WSN’s transmission. Therefore, how to suppress or eliminate the negative influence exerted by external malicious jamming on the signal transmissions among its sensor nodes is a very important issue in the research and optimal design of WSNs. Traditional anti-jamming communication methods, which are based on extended spectrum technology, can be used to deal with routine jamming effectively. They mainly include frequency hopping spectrum expansion (FHSS) [3] and direct sequence spread spectrum (DSSS) [4]. However, with the development of intelligent jamming methods, those traditional anti-jamming communication schemes are beginning to show their many weaknesses in ensuring the reliability and effectiveness of the communications among different sensors in a WSN. Moreover, with the expansion of a WSN, the mutual interference among its sensor nodes increases, which should also be taken seriously to ensure its performance.

Machine learning, which can be regarded as a technique for the machines to learn from the collected data to optimize their own decision making [5], provides a feasible new way to cope with intelligent malicious jamming [6,7]. What is more, reinforcement learning, a vital branch of machine learning, is capable of ensuring anti-jamming communications based on joint actions executed by the WSN and the feedback from the external environment, without the need for jamming modeling [8]. Anti-jamming communication schemes based on reinforcement learning have already been extensively studied [2,7,9,10,11,12,13,14,15,16,17]. In reference [7], a SARSA-based anti-jamming algorithm was proposed to counter random pulse jamming in a time domain; however, most of the current research focuses on interference patterns with frequency domain characteristics. In references [9,10], an anti-jamming scheme based on Q-learning is presented, which is widely regarded as a classic scheme without the need of jamming modeling. However, it only considered the single-agent network scenario. A wireless relay node (WRN) with multiple agents was considered in references [2,11]. However, in these studies, the jamming modeling was still based on the traditional sweep jamming, which shows a lot of regularity and can be easily learned and mastered by a WSN. The jamming model used in reference [12] features pseudorandom characteristics, which can be regarded as a kind of intelligent jamming. In reference [13], a Dyna-Q-based algorithm was proposed to obtain better adaptability and faster convergence speed by updating the value function and adjusting the external environment model in real time, while the external environment model is fixed in the classical Q-learning process. With the alternating development of jamming and anti-jamming research, it has become an established trend to explore new technologies and possibilities to improve the performance and intelligence of the anti-jamming schemes for a WSN in different application scenarios. For example, in reference [15], an actor–critic algorithm was mentioned as a promising algorithm for improving the anti-jamming performance of the communications among WSN sensor nodes. However, it did not present any specific scheme. Reference [16] proposed an actor–critic-based anti-jamming communication algorithm for a single-agent system. Reference [17] proposed an actor–critic-based algorithm for a multi-agent system without jamming, and the simulation results revealed that the actor–critic-based scheme has a greater convergence speed than the traditional schemes based on the ability of learning from collected data from the environment. 

By injecting malicious jamming into the wireless channel, jammers can seriously reduce the performance of wireless sensor networks or even interrupt the transmission [18,19]. In addition, the insufficient interaction between sensors in the network will easily lead to competition in the same channel, which will lead to failure of transmission and affect transmission performance badly. It is necessary to consider both internal and external factors that affect the transmission performance of wireless sensor networks to optimize the efficiency of the system. 

In this paper, an actor–critic-based scheme for a WSN with distributed multi-agent systems is proposed which can improve the anti-jamming communication performance among wireless sensors in the environment of intelligent malicious jamming for WSNs. The proposed scheme increases the system’s operation efficiency by dividing the system’s strategy evaluation and execution into two independent parts, when dealing with external jamming and internal interference. The simulation shows that the proposed scheme can improve the efficiency of communications among the network nodes and has a better ability to learn from internal and external environments, resulting in faster algorithm convergence. 

The rest of the paper is organized as follows: Section 2 presents the system model and necessary assumptions. Section 3 introduces the Distributed Anti-Jamming Algorithm (DAJA) based on an actor–critic scheme. Section 4 presents the simulation analysis results and discussions, and Section 5 gives the concluding remarks. 

## 2. Assumptions and System Model

### 2.1. Assumptions of the System

For simplicity, the following assumptions are made in this study:

There are N active sensors in this wireless sensor network system and M communication channels that are available to the sensor nodes. In a typical multi-agent communication system, sufficient channels are always provided, so we assume N<M tacitly. The schematic diagram of the system is shown as Figure 1.We assume that malicious jamming exists in the multi-agent system, according to reference [17], which represents the real-life application environment. In addition, in this system, we assume that the real-time information exchanges among the sensors are not possible, so a network sensor does not know the channels used by other sensors. That is to say, channel competition among all the sensor nodes is inevitable. We define that the competition happens when more than one sensor tries to occupy the same channel at the same time. When competition happens, the involved sensor nodes cannot transmit data successfully. In addition, we define that the effect of malicious jamming conducted by the jammer on all the sensors in the system is exactly the same, regardless of the jammer and sensors’ positions. In addition, we assume that the channel noise is not prominent enough to affect the data transmission among all the sensors.We assume that the data transmission among the sensor nodes happens in a series of timeslots. A timeslot has a length of $T_{s}$, which is the minimum time unit for data transmission. Malicious jamming also happens in a series of timeslots of this kind. A timeslot can be divided into two parts: a perception part $T_{observe}$ and an action part $\;T_{act}$. Figure 2 shows the structure of a timeslot. In the $T_{observe}$ part of a timeslot, sensors in the system can sense and analyze the channel where malicious jamming might be present. At this stage, a sensor node can perform anti-jamming algorithms based on its analysis and choose the best channel for data transmission. In the $T_{act}$ part of a timeslot, the node can transmit data. In a $T_{act}$, each node can only occupy one channel for data transmission, but in a $T_{observe}$, each node can sense and analyze all the available channels.This paper assumes that the malicious jamming fabricated by the enemy entities is high-power jamming in unit timeslot $T_{s}$, which can interrupt and even intercept a sensor’s communication instantly and result in transmission failure. Moreover, all the sensors in the WSN are in jamming’s reach. In order to keep the environment the WSN is working in stable, there is no resources depletion for the malicious jammer from the enemy, which means the jammer can always work and the threat of the malicious jamming is always there. The enemy jammer can sense all the channels used by every node and can select J channels with the highest utilization rate at the current stage to exert probabilistic jamming. A single jamming lasts L timeslots. However, compared with intelligent jamming assumptions used in reference [12], the jamming model used in this paper can change their jamming parameter periodically, which is more intelligent and can be more threatening to the communications among the sensor nodes. In this paper, we define this jamming pattern as multi-channel probabilistic tracking jamming (MPT-jamming).We assume all the sensors in the system execute all actions synchronously, including channel sensing and data transmission. All the sensor nodes have the same sensing ability and can obtain the same sensing results under the same conditions. Moreover, for simplicity, we assume that the enemy jammer’s working time is synchronous with the nodes’ communication time. They both work in the same timeslots.

### 2.2. System Model and Problem Formula

This paper models the problem of wireless sensors networks anti-jamming communication as a Multi-Agent Markov Decision Process (MAMDP). The definition of the five elements 〈N,S,A,P,R〉 in the MAMDP is shown as follows: N is the number of sensors in this system, S is the state space, A is the action space, P is the state transition probability, and R is the immediate reward that the transmitters can receive from the system. The relevant formula in this section is based on reference [12].

S: The state space mainly indicates the basic situation of communication channels. Take channel c as an example. When c is idle, the channel state is marked with id; if c is occupied by jamming, c will be marked with ja. The sensor’s decision is based on whether each channel is idle or not, and the channels’ states are the knowledge that the agents need to sense. The state space is defined as follows:(1)$$S≜\left\{s|s=\left(b_{1}\cdots b_{Z}\right)\right\}$$
where $s=\left(b_{1},b_{2}\cdots b_{Z}\right)$ represents the collection of jammed channels observed by sensors, subscripts of $b_{1}\cdots b_{Z}$ indicate the initial channel number, sorted from smallest to largest; Z∈[1,M] and Z∈R. So, the state space S in this paper has $C_{M}^{Z}$ possible states. A: In a multi-agent system, action space is a joint action set composed of independent actions selected by various agents. Here, we set every sensor in this system to only select one channel from M channels provided when transmitting. Therefore, the independent action sub-space $A_{i}$ of a single communication sensor in action space A is the same, $A_{1}=A_{2}\cdots =A_{N}$. The definition of the sub-space of action belonging to each sensor is as follows: (2)$$A_{i}≜\left\{a_{i}|a_{i}\in \left\{1,2\cdots M\right\}\right\}$$
where i∈{1,2⋯N} is used to distinguish sensors, and $a_{i}\in \left\{1,2\cdots M\right\}$ represents the channel selected by sensor i. So, combine all the sub-space, and the joint action a is the combination of transmission channels of each sensor: (3)$$a=\left\{a_{1},a_{2}\cdots a_{N}\right\}$$The definition of joint action space A is as follows: (4)$$A=A_{1}⊗A_{2}⊗\cdots ⊗A_{N}$$
where ⊗ represents the Cartesian product. In addition, according to the above settings, a single communication sensor can select only one channel for transmission at a time; in that way, a single sub-space $A_{i}$ has M possible action, and joint action space A has $M^{N}$.

3.P: $S\times A\times S^{\prime }\mapsto \left[0,1\right]$ represents the probability of all the sensors transmitting the state from S to $S^{\prime }$ by conducting joint action A.4.R: In general, the reward obtained by sensor i conducting sub-action $a_{i}$ in state s depends on whether there are other communication sensors or jamming in the same channel. Figure 3 takes N=3,M=4 as examples, showing the schematic diagram of the transmission relationship among sensors by cube without jamming from the perspective of communication sensor 1.

As shown in Figure 3, three sensors constitute three coordinate axes of three-dimensional space, respectively, and the serial number of channels is the scale of each coordinate axis. From the perspective of sensor 1, the green cube indicates that there is no channel competition with other sensors, so the reward of successful transmission $r_{1}\left(s,a_{1}\right)$ will be rewarded immediately. If there is more than one sensor occupying one channel to transmit, the red cube indicates the transmission is a failure. Figure 4 shows the other sensors’ relationship with sensor 1 when it is transmitting in different vision according to the diverse selection of it. 

To this end, the definition of the immediate reward function for every single sensor’s successful transmission is as follows: (5)$$r_{n}\left(s,a_{n}\right)=\{\begin{array}{c} 1\;\;\;\;\;\;\;\;\;\;a_{n}\neq j_{z},a_{n}\neq a_{m}\\ \;\;\;0\;\;\;\;\;\;\;\;\;\;\mathrm{else}\;\;\;\;\;\;\;\;\;\;\;\;\;\;\;\;\;\;\;\;\;\;\;\; \end{array}$$
where m∈{1,2⋯N} and n≠m, and the immediate reward for successful transmission in one timeslot by a single communication sensor is defined as 1. According to the settings above, all sensors in the system share the instant reward function, the reward function of the system takes joint actions, and $a=\left\{a_{1},a_{2}\cdots a_{N}\right\}$ is the sum of immediate reward (Equation (5)) for a single sensor: (6)$$R_{i}\left(s,a\right)=\sum_{n=1}^{N}r_{n}\left(s,a_{n}\right)$$

## 3. Distributed Anti-Jamming Algorithm (DAJA) Based on Actor–Critic

An actor–critic algorithm does not apply conventional value function iterative algorithms alone, such as Q-learning; it focuses on solving the problem of strategy selection updating the efficiency in algorithm rounds. An A–C algorithm can realize a real-time update in the process of decision iteration: value function is parameterized and estimated by the critic part, while the actor part guides the real-time update of strategy according to the value function estimated by the critic part. Its structural flow diagram is shown in Figure 5, where the solid box line represents two interactive subjects in reinforcement learning: environment and system. The system is composed of multiple agents in this paper. The dotted box line in Figure 5 represents the two sub-modules of system decision, actor and critic, which are responsible for the policy selection and evaluation of the system, respectively. In short, the A–C algorithm optimizes the decision-making process and improves the decision-making efficiency by separating the evaluation decision and execution decision in the decision-making process and making them independent. In addition, the real-time update strategy meets the requirements of multi-agent real-time anti-intelligent interference and ensures timeliness. 

### 3.1. The Critic Part

According to the distributed multi-agent scenario, the critic evaluates the strategy by means of time-difference error (TD error) using parameter approximation. In this paper, multi-agent SARSA (MAS) is used to iterate the evaluations of the strategy by incremental linear approximation of state-action value functions. The definition of updates is as follows: (7)$$Q\left(s_{t},a_{t}\right)=\overset{N}{\underset{n=1}{\sum }}\left\{Q_{n}\left(s_{t},a_{t}\right)+\alpha \left\{r_{i,t}+\gamma \left[Q_{n}\left(s_{t+1},a_{t+1}\right)-Q_{n}\left(s_{t},a_{t}\right)\right]\right\}\right\}$$
where α is the learning parameter, γ is the discounted parameter, $s_{t}$ is the joint state of all the sensors in the system at time t, and $a_{t}$ is the joint action users conducted at time t.

By introducing the approximate parameter ω, the approximate state-action value function $\hat{Q}\left(s_{t},a_{t},\omega \right)$ is as follows:(8)$$Q\left(s_{t},a_{t}\right)=\hat{Q}\left(s_{t},a_{t},\omega \right)$$

The method of parameter approximation is to express the value function (in this paper, the state-action value function, i.e., *Q* value) as a linear combination of value functions, and the definition is as follows: (9)$$\hat{Q}\left(s_{t},a_{t},\omega \right)=\omega^{T}Q\left(s_{t},a_{t}\right)=\overset{N}{\underset{i=1}{\sum }}\omega_{i}Q_{i}\left(s_{t},a_{t}\right)$$
where the number of users N set in this paper is the dimension of the linear combination, state-action space, at this time. It is worth mentioning that the optional channel M is the range of the state-action space. For example, taking N=3, M=10 as examples, the state-action pair is a three-dimensional space in the range of 10×10×10. 

The definition of existing TD error $\delta_{t}$ is as follows:(10)$$\delta_{t}=R_{t+1}+\gamma \omega^{T}Q\left(s_{t+1},a_{t+1}\right)-\omega^{T}Q\left(s_{t+1},a_{t+1}\right)$$
where $R_{t+1}$ is the reward all the users cumulatively received from the environment at the next timeslot t+1.

The definition of ω is as follows:(11)$$\omega_{t+1}=\omega_{t}+\beta \delta_{t}Q\left(s_{t},a_{t}\right)$$
where β is the other discounted parameter. 

### 3.2. The Actor Part

In this paper, the part of actor employs stochastic policy algorithms (SPA). In order to ensure the timeliness of the communication anti-jamming decision, real-time update of on-policy is adopted at the same time. 

The discrete random strategy $\pi_{\theta }\left(a|s\right)$ for the system at time t satisfies the following distribution: (12)$$\pi_{\theta }\left(a_{t}|s_{t}\right)=Pr\left(a_{t}|s_{t};\theta \right)$$
where θ is the strategy parameter. Because of the discrete joint action space, leading to the discretization of the joint strategy, the policy distribution $Pr\left(a_{t}|s_{t};\theta \right)$ exports by SoftMax, so the random strategy is defined as follows: (13)$$\pi_{\theta }\left(a_{t}|s_{t}\right)=e^{Q_{\theta }\left(a_{t}|s_{t};\pi \right)}/\overset{M}{\underset{i=1}{\sum }}\overset{N}{\underset{i=1}{\sum }}e^{Q_{ij}\left(s_{t}\right)}$$
where $Q\left(a_{t}|s_{t};\theta \right)$ represents users conducting joint action $a_{t}$ by selecting strategy π at time t when they are in state $s_{t}$. In short, the definition above is the result of introducing strategy parameter θ in $Q\left(s_{t},a_{t}\right)$, and the value remains. $Q_{ij}\left(s_{t}\right)$ represents time t, with user i independent cation selecting channel j when in joint state $s_{t}$.

### 3.3. The Proposed DAJA Algorithm

According to the process block diagram of the algorithm shown in Figure 5 and the parts of critic and actor described in the previous two sections, the proposed DAJA algorithm can be described as follow:

All users in the system conduct the joint action $a_{t}$ according to the policy selection result of the previous timeslot and receive the immediate reward $R_{t+1}$ accumulatively at the same time. On this basis, the system forecasts the joint state $s_{t+1}$ in the next timeslot for all the users involved. Then, the joint strategy is updated according to Equation (13), and the joint action $a_{t+1}$ of the system is predictable. TD error $\delta_{t}$ and approximate parameter ω will be updated separately according to Equations (10) and (11) until ω convergence. At this point, we believe that the optimal strategy $\pi^{*}$ is obtained, that is, the realization of a multi-agent anti-jamming decision. 

The steps of the DAJA (Algorithm 1) are as follows:
**Algorithm 1**: DAJA1: **Initialize**: α, β, γ, θ, s∈S, a∈A, Q(s,a)←0;2: **for** t=1,2⋯T
**do**3:  Users conduct sub-action $a_{i}$ according to the joint action a made in last timeslot.4:  System senses the environment and receives joint state s from the external environment, and the jammed channels are sorted according to Equation (1).5:  The sub-module critic part of the system starts to work, and users determine whether there is competition or jamming in each channel.6:  Users calculate their respective reward, respectively, by Equation (5), and the joint reward R will be accumulated by Equation (6).7:  System forecast the next joint state $s^{\prime }$ by current action a.8:  The sub-module actor part of the system starts to work, and the system follows the strategy $\pi_{\theta }$ to select joint action $a^{\prime }$ in next timeslot.9:  TD error $\delta_{t}$ and approximate parameter ω will be updated separately by Equations (10) and (11).10:   Update joint state s and joint action a: $s\leftarrow s^{\prime }$, $a\leftarrow a^{\prime }$ and output them to next iteration.11:   t=t+112: **end for**

After the completion of initialization, the algorithm repeats the following operations in each full timeslot: at the very beginning of every timeslot, users in the system conduct sub-cation $a_{i}$ by the decision a made in the last timeslot (line 3). In sub-slot $T_{observe}$, the system senses the external environment and receives the current joint state s (line 4). The jammed channels will be sorted according to Equation (1). At this time, the sub-module critic part of the system starts to work. By integrating the information obtained in the first two steps, users determine whether there is competition or malicious jamming in channels, respectively, (line 5) so that every user involved could calculate their own immediate reward from the environment by Equation (5) and accumulate the result, that is, the immediate reward of the system R based on Equation (6) (line 6). In sub-slot $T_{act}$, the system forecasts the next state $s^{\prime }$ based on current action a (line 7). Then, the sub-module actor part of the system goes into operation, and the system follows the strategy $\pi_{\theta }$, which updates when information from the environment about state sends in the last timeslot, to select joint action $a^{\prime }$ in the next timeslot (line 8). TD error $\delta_{t}$ and approximate parameter ω will be updated separately by Equations (10) and (11) (line 9). Finally, the iteration in a timeslot ends with next state $s^{\prime }$ and action $a^{\prime }$ being sent back to the system (line 10). The cycle repeats until the end of the iteration. Combined with Figure 5, the steps of DAJA based on actor–critic are shown in Figure 6.

Each user in the system repeats the above steps in each timeslot before the transmission terminates. After a finite number of iterations, when the approximate parameters ω are convergent, in this timeslot, the DAJA also converges to the optimal strategy $\pi^{*}$. We consider that, in this moment, DAJA approaches the convergence. 

In conclusion, the proposed DAJA algorithm has the capacity to realize strategy evaluation by using the multi-agent SARSA learning method to linearly approximate the current real state-action value function and finally succeed in multi-agent communication anti-jamming decision by using the online learning SoftMax discrete strategy method to ensure timeliness. 

## 4. Simulation Results and Analysis

### 4.1. Parameter Settings

In order to evaluate the performance of the DAJA algorithm applied for anti-intelligent jamming communication in wireless sensor networks, we adopt the method of control variable in this simulation. Except for the newly proposed structure of the timeslot and the parameters in the A–C algorithm, other parameters, e.g., the number of available channels M, sensors, namely the agents in the system N etc., are consistent with those in the previous work [12] of our team.

We set the parameters related to the simulation in Table 1 as follows:

In order to further evaluate the performance of DAJA proposed in this paper, the following two anti-intelligent jamming schemes for WSN, including a reinforcement learning-based scheme and a traditional scheme, are set up for comparison: Joint multi-agent anti-jamming algorithm (JMAA) [12]: multi-agent Q-learning is used to realize knowledge sharing by exchanging Q-tables in the anti-jamming decision, so as to achieve anti-jamming channel selection online.Orthogonal frequency hopping (OFH): each sensor switches the transmission channel according to the randomly generated fixed frequency hopping pattern, and the frequency hopping pattern of different sensors is orthogonal to each other to ensure that the same channel is occupied by only one user at the same time.

To compare the transmission performance and convergence speed of algorithms mentioned above effectively, the following two performance indicators are set: 

Average Reward of Transmission in Each Cycle (ARTEC). Firstly, we define L timeslots, as long as jamming’s duration, as a transmission cycle, so the duration is $L\times T_{s}$ naturally. The reward of successful transmission in one cycle is the successful timeslot $R_{total}=\sum R_{i}$, where i is used to distinguish users, the single reward of one user is $R_{i}=E\cdot \tau_{trans}^{W}\left(k\right)$, and $\tau_{trans}^{W}\left(k\right)$ is the number of timeslots, where user i transmits successfully in one cycle, and the number of cycles is $W=N_{S}/L=2000$, k∈{1,2,3⋯W}. To describe transmission performance, we further define average transmission reward $R_{average}=\sum R_{i}/k$, which is used to describe the average transmission capacity of the algorithm after the kth cycle. Obviously, ARTEC represents the ability of system’s transmission when facing intelligent malicious jamming, and the higher the value, the better the transmission. Normalized Velocity of Learning Anti-jamming (NVLA). In this paper, we define $v=W/k_{submit}$, where W is the number of the transmission cycles set above. According to the parameter mentioned above, in this paper, $W=N_{s}/L=2000$, and $k_{submit}$ is the cycle’s number that remains stable after the algorithm reaches the peak value of the system’s transmission reward, that is, the system applies the algorithm to achieve convergence. The definition of v is used to describe how fast the algorithm learns external disturbances; the higher the value, the faster the system becomes convergent. The relative speed of different algorithms is the ratio of the speed of two algorithms; for instance, the relative speed between DAJA and JMAA is the ratio of $v_{DACA}$ and $v_{JMAA}$ $v_{comp-D\&J}=v_{DACA}/v_{JMAA}$, which is also equal to the inverse ratio of the number of transmission cycles reaching their respective theoretical reward peak $v_{comp-D\&J}=k_{submit-J}/v_{submit-D}$.

### 4.2. Simulation and Analysis

The algorithm proposed is simulated using MATLAB. 

According to the parameter set above, the cubic diagram of timeslot-channel selection at the very beginning of the simulation (within the 0–30 timeslot) is obtained, as shown in Figure 7.

Since the MPT-jamming set in this paper is the probabilistic jamming based on the channel selection of sensors in the previous timeslot, the real-time probability distribution diagram of timeslot-channel jamming is drawn for the 0–30 timeslot in Figure 8 according to Figure 7.

Combined with the probability distribution of intelligent jamming and persistence of its settings in this paper, the real-time communication-jamming diagram, within the initial 30 timeslots, is as shown in Figure 9. 

To keep the length of time of the system’s convergent state consistent with that of the initial state, timeslots from 2000 to 2030 are selected for comparison. The real-time diagram of transmission jamming is shown in Figure 10 when the system is convergent.

According to Figure 10, the algorithm proposed for WSNs can realize anti-jamming communication applied to a multi-agent system. Compared with JMAA proposed in reference [12], each sensor in the system only occupies two channels for convergence, which can be regarded as one primary channel and the other as the standby; here, this paper defines this policy for anti-jamming communication as the ‘One primary, One Standby’ policy. Taking sensor 1 in Figure 10 as an example, the primary channel of sensor 1 is one and the alternative is two. It is worth mentioning that, according to the system settings in this paper, the channel selection of MPT-jamming is based on the channel selected by sensors in a previous timeslot. The intelligent jammer executes MPT-jamming in the next timeslot, lasting five timeslots, so that there is one timeslot difference existing between jamming’s selection and execution naturally. We define the MPT-jamming itself as hysteresis. This phenomenon will affect the application of the algorithm proposed, which will be discussed in more detail later.

In Figure 10, when the system is convergent, sensor 1′s transmission without jamming can be realized by implementing the ‘One Primary, One Standby’ policy, which is the same for the other sensors in system. Compared with reference [12], when JMAA converges, each sensor occupies tree channels and nine channels in total, and the channel occupancy rate decreases when putting DAJA into practice. 

After verifying the feasibility of the proposed algorithm in multi-agent anti-intelligent jamming for the WSN, we evaluate its performance further by comparing the performance of transmission and speed of convergence of the proposed algorithm with the existing anti-jamming scheme JMAA [12] based on reinforcement learning and conventional scheme OFH in the same environment. 

Figure 11 shows that the transmission performance of DAJA is better than that of JMAA and significantly better than OFH’s performance. Compared with the final value, DAJA is 0.800, which is the best of three scheme for the WSN, while JMAA is 0.747 and OFH is 0.6. The scheme based on actor–critic proposed in this paper is 7% higher than that of the existing scheme based on multi-agent joint Q-learning, while being 33% higher than the conventional scheme. In terms of transmission, it is obviously shown that DAJA has the best performance among the three schemes involved in this paper when applied to a WSN countering intelligent malicious jamming. As the model-free reinforcement learning, there is also difference in the iteration strategy between Q-learning and SARSA. In the previous work, our team has clarified the difference that Q-learning tends toward aggressive strategies, striving for high rewards from a high-risk environment, and SARSA favors a conservative strategy, which leads to the difference in performance comparison [7]. Taking a closer look at Figure 10, we find that the traditional anti-intelligent jamming scheme OFH can respond quickly and ensure the stability of the multi-agent system, but the cost of rapid convergent is that the transmission performance has finiteness. OFH’s transmission has its finiteness because this traditional anti-jamming scheme can avoid interference between sensors in the networks but cannot avoid external malicious jamming completely. Moreover, because of the difference between JMAA and CAJA in the action strategy, there will be too many conflicts between jamming and JMAA’s transmission; in this case, the system’s transmission reward is not as good as CAJA’s. The decision of anti-jamming communication not only requires the system to converge as soon as possible, but high-quality transmission also matters, which is a significant guarantee that the system can operate properly, especially in an intelligent jamming environment for the WSN. This is one of the main reasons why reinforcement learning is selected to design anti-intelligent jamming communication decisions for WSNs. In the jamming environment set in this paper, multi-agent systems do not act in unison, and there is not only cooperation but also competition existing among sensors. To the sensors in the system, they need to learn from the external and internal environment to ensure high-quality transmission. The use of reinforcement learning, a trial-and-error theory, is convenient for each sensor to regard the other sensors as the internal environment, avoiding the competition. In order to realize high-quality transmission under the threat of intelligent malicious jamming for a WSN, a scheme based on reinforcement learning is effective. As can be seen from Figure 11, the anti-jamming scheme based on reinforcement learning, rather than a traditional scheme, can adapt to intelligent jamming and achieve high-quality transmission; in that case, the scheme proposed performs even better than the existing scheme in terms of transmission.

In order to compare the convergence speed of DAJA, JMAA, and OFH, we use the table method to select the rewards of each timeslots period in the pre-convergence state of the three schemes to compare the convergence speed. 

It can be seen from Table 2 that at the first timeslot period, namely 407~403 cycle of the timeslot, DAJA converges completely, the system has maintained reward 15 in the following cycles, and JMAA converges within the timeslot in the second period. As can be seen from Figure 11, after the initial iteration, the system applying OFH reaches convergence quickly at the cost of transmission performance. As the conventional anti-jamming scheme, OFH’s strategy is fixed and independent of the external environment, which means OFH could converge with ordinary transmission performance at the beginning, no matter whether the jamming is intelligent or not. However, the reinforcement learning-based anti-jamming schemes, DAJA and JMAA, adapt to the environment gradually by the immediate rewards through the interaction with the environment and finally approach anti-jamming communication, which is determined by the characteristic of reinforcement learning algorithm. DAJA converges faster than JMAA because of conservative strategy, which means JMAA is prone to transmission failure because of its aggressive strategy under high risk, resulting in longer time than DAJA to adapt the environment and converge. According to the index, the convergence speed of DAJA is about 4.94, and that of JMAA is 4.38; we also regard that the OFH’s has no value of reference. In horizontal comparison, the relative speed ratio between DAJA and JMAA was 1.13, which increased by 12.84%, showing the better performance in convergence. The pairwise relative speeds are shown in Table 3.

As in-depth research of reference [12], we (all the authors in this team) are going to focus on the comparison of performance among the scheme we proposed (DAJA) and the existing joint Q-learning-based scheme (JMAA [12]) and conventional scheme (OFH) in the same environment, including the same parameter values in the system and the same intelligent malicious jamming. Through the control variable method and the transitivity of performance differences in the simulation process, the simulation shows that the DAJA’s performance of transmission and speed of convergence in the environment we set are better than that of JMAA. According to the main purpose of our work, which is to achieve reliable transmission under the threat of intelligent malicious jamming, higher-quality transmission performance is much more important than faster convergence, because we believe that the difference in convergence speed can be reduced with sufficient computational power. Therefore, we conclude that the proposed DAJA is better than the joint Q-learning-based scheme and traditional scheme when applying to a multi-agent system countering intelligent malicious jamming for a WSN naturally. 

It should be pointed out that, according to the simulation results and the difference in iterations between the two reinforcement, learning-based anti-jamming algorithms, we conclude that, unlike DAJA, the actor–critic-based algorithm can update strategy evaluation and action selection in one iteration simultaneously. JMAA and the Q-learning-based algorithm only update the strategy in one iteration and then guide the action making. The ‘One iteration, Two updates’ strategy DAJA applied makes it more adaptable to the environment, with high benefits of more accurate action making in one iteration, leading to the faster convergence. JMAA has slower convergence than DAJA because it only updates the strategy evaluation part, the so-called critic part, in one iteration, which means it takes more time to adapt to the environment, resulting in worse performance in convergence and transmission.

### 4.3. Further Analysis and Future Work

The simulation results show that DAJA is superior to JMAA in terms of transmission and convergence applying to a multi-agent system, and the system’s saving channel resources under the threat of intelligent malicious jamming set in this paper mean a lot but not everything. On the contrary, the saved channels constrain the application of DAJA in some specific conditions. The reason is the jammer and multi-agent system execute actions synchronously. In the case of time delay error existing between jamming and transmission, the real-time cubic diagram of the system’s initial state is as shown in Figure 12. The ‘One Primary, One Standby’ strategy of the multi-agent selected will have a negative impact on the transmission of the system overall, as shown in Figure 13. 

After executing the ‘One Primary, One Standby’ strategy, the sensors in the WSN will be affected by the tail of jamming implemented, and the reason is the time delay error between the jammer and the system. When the system converges, the theoretical peak value of transmission reward within one timeslot cycle will be reduced, and it will be reduced by 20% in the environment set in this paper. 

In contrast, JMAA, due to the more alternative channels for sensors in system to select, will not be affected by the jamming’s tail as easily as DAJA. In addition, it inspires us that when the provided channel is limited, reducing the channel utilization rate is not foolproof. Making the most of the channel resources under a specific environment will provide a significant guarantee for the transmission with high quality. 

For our future research, we will expand the application of DAJA in more environments and enhance its applicability under the premise of improved transmission performance and decision-making learning speed. Before we test the effectiveness of the proposed algorithm in practical applications, more practical details need to be taken into consideration, e.g., the position of the jammer and the layout of the sensors in the WSN. In addition, due to the proposed algorithm still being in the theoretical stage so far, the performance of DAJA implemented in real hardware remains to be seen. This will be our future work: to test the performance of the proposed algorithm when implemented in hardware as well.

## 5. Conclusions

In this paper, we proposed a Distributed Anti-Jamming Algorithm (DAJA) based on an actor–critic scheme for a WSN. The proposed algorithm models the multi-agent communication as a Multi-Agent Markov Decision Process (MAMDP) when facing intelligent malicious jamming. In order to optimize the decision-making structure and improve the decision-making efficiency of the multi-agent system, making the evaluation and execution of the system’s strategy divided is considered promising in accomplishing anti-intelligent communication effectively. By using the visualization method of a transmission cube and superimposing the thermal map of intelligent jamming probability distribution, the real situation of anti-intelligent jamming communication of a multi-agent system is displayed, even though the channels’ resource is finite. In addition, DAJA is more efficient in iteration with two updates of policy evaluation and action selection. At this time, the system’s action selected is more accurate after optimizing, and the ability to adapt to the external environment is better than existing anti-jamming schemes, resulting in better performance in transmission and convergence. The simulation shows that DAJA achieves the anticipated effect. By further comparison of transmission and convergence, we found that a DAJA-based scheme can outperform existing joint Q-learning-based schemes and tradition schemes in terms of transmission and convergence for a WSN. The simulation results show that, compared with JMAA [12], the proposed algorithm improves the WSN’s transmission rewards by 7% and convergence speed by 12.84%. 

## Figures and Tables

**Figure 1 sensors-22-08159-f001:**
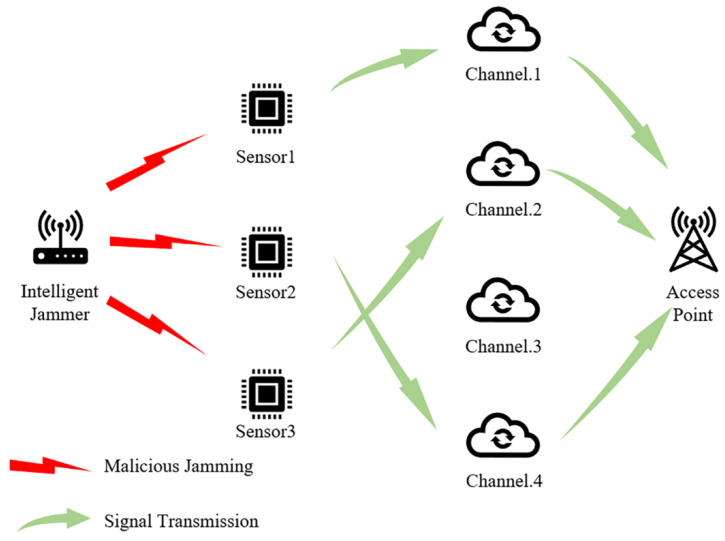
The schematic diagram of system model (take N=3,M=4 as example).

**Figure 2 sensors-22-08159-f002:**
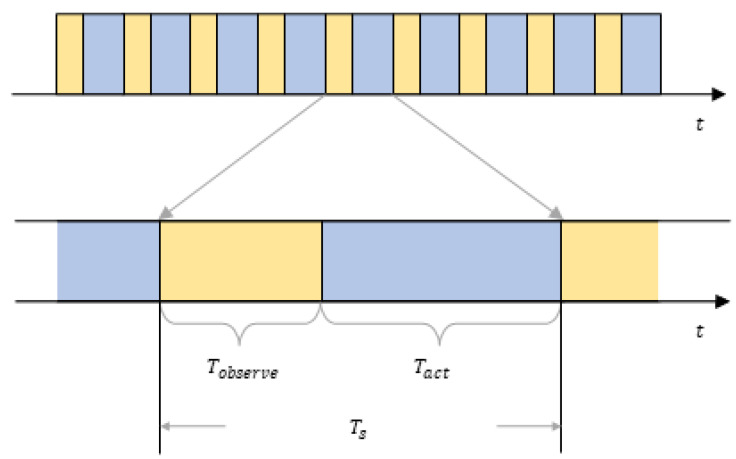
The schematic diagram of single timeslot’s structure.

**Figure 3 sensors-22-08159-f003:**
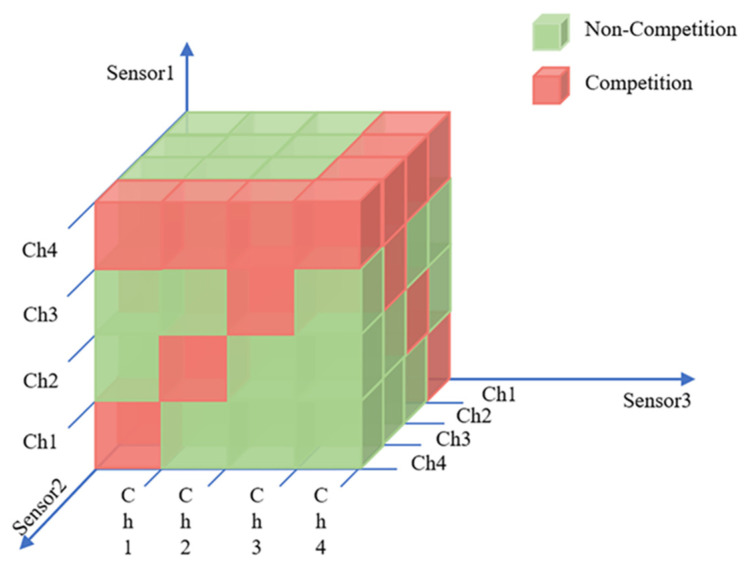
The cubic diagram of the relationship among sensors.

**Figure 4 sensors-22-08159-f004:**
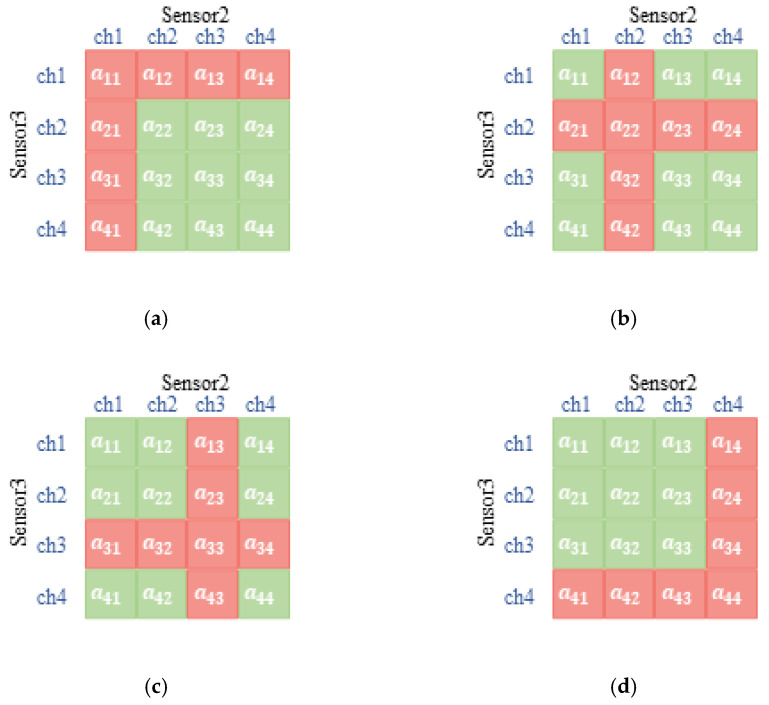
The diagram of the cubic relationship among sensors in sensor 1′s version. (**a**) Sensor 1 chooses channel 1; (**b**) sensor 1 chooses channel 2; (**c**) sensor 1 chooses channel 3; (**d**) sensor 1 chooses channel 4.

**Figure 5 sensors-22-08159-f005:**
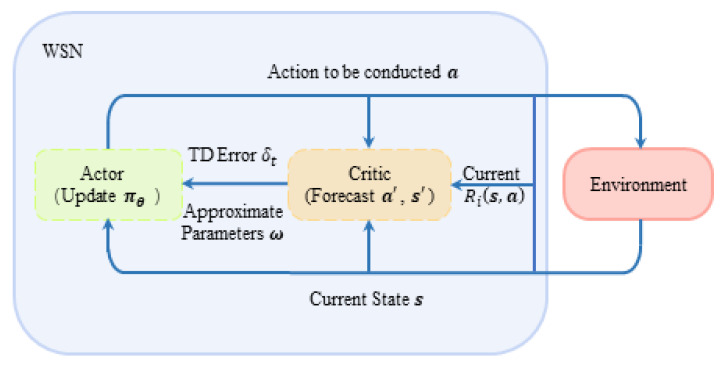
The diagram of DAJA’s structure applied to multi-agent system.

**Figure 6 sensors-22-08159-f006:**
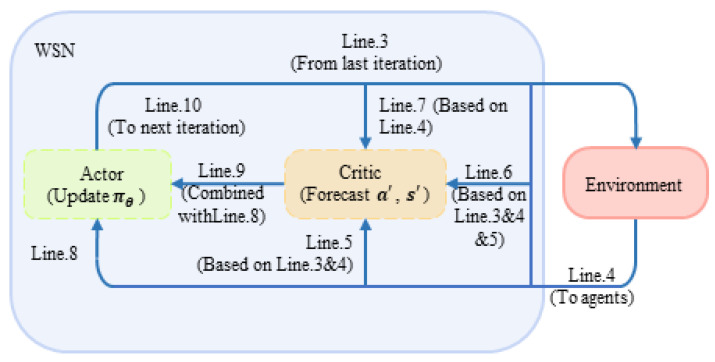
The diagram of DAJA’s flow based on actor–critic.

**Figure 7 sensors-22-08159-f007:**
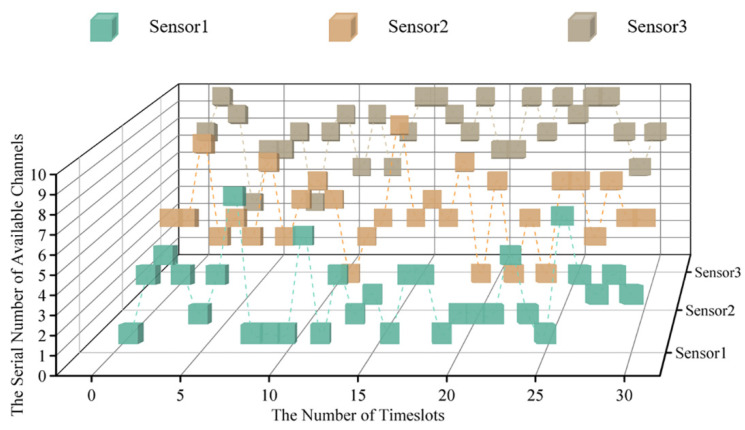
The cubic diagram of timeslot-channel selection in 3 sensor systems.

**Figure 8 sensors-22-08159-f008:**
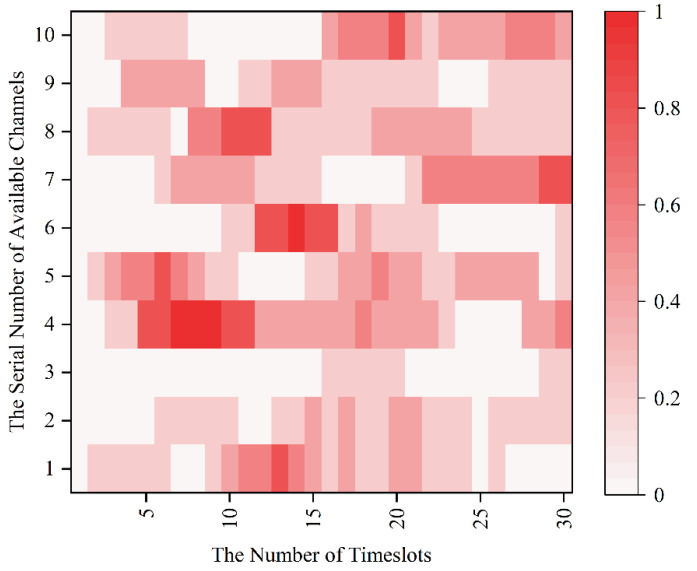
The real-time probability distribution diagram of the of timeslot-channel jamming.

**Figure 9 sensors-22-08159-f009:**
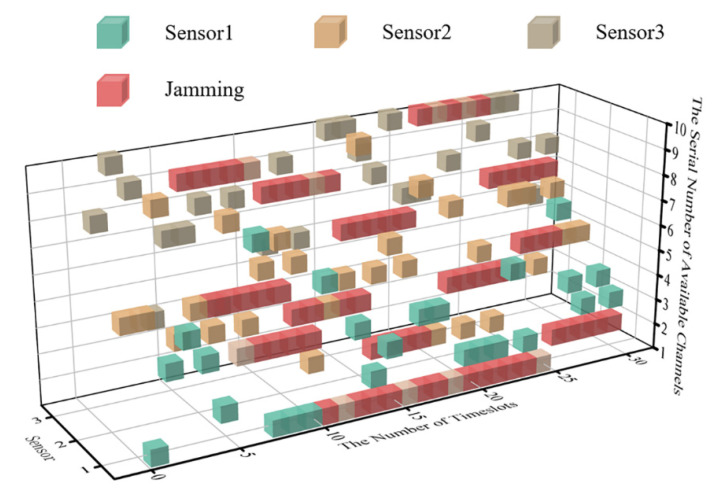
The cubic diagram of DAJA’s initial state.

**Figure 10 sensors-22-08159-f010:**
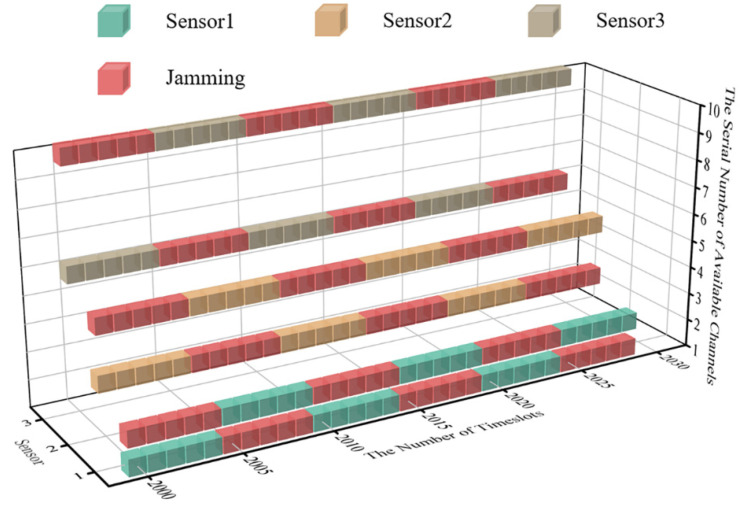
The cubic diagram of DAJA’s convergent state.

**Figure 11 sensors-22-08159-f011:**
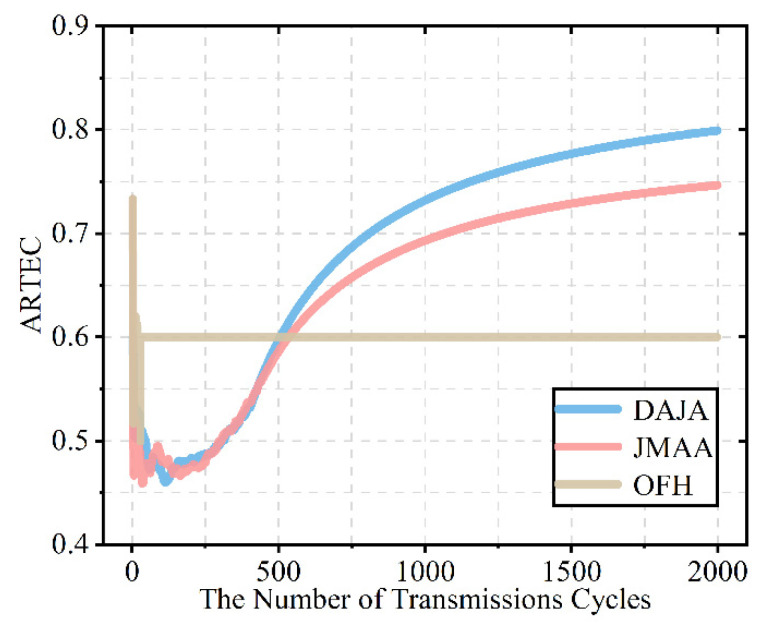
The schematic diagram of transmission performance comparison among DAJA, JMAA [12], and OFH.

**Figure 12 sensors-22-08159-f012:**
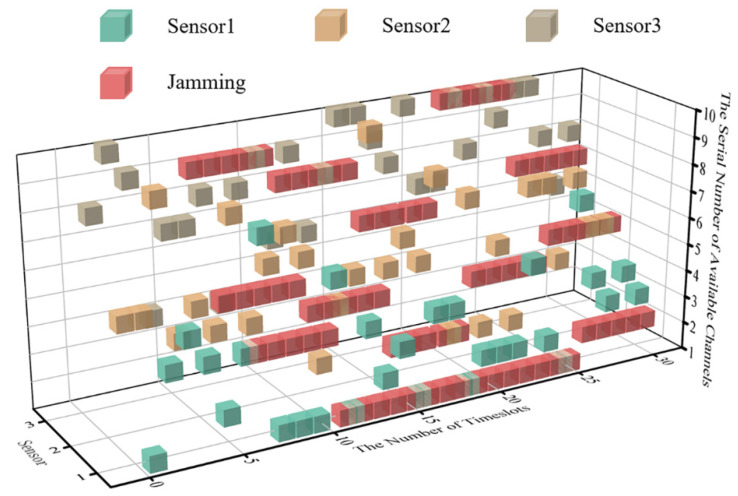
The cubic diagram of DAJA’s initial state with time delay.

**Figure 13 sensors-22-08159-f013:**
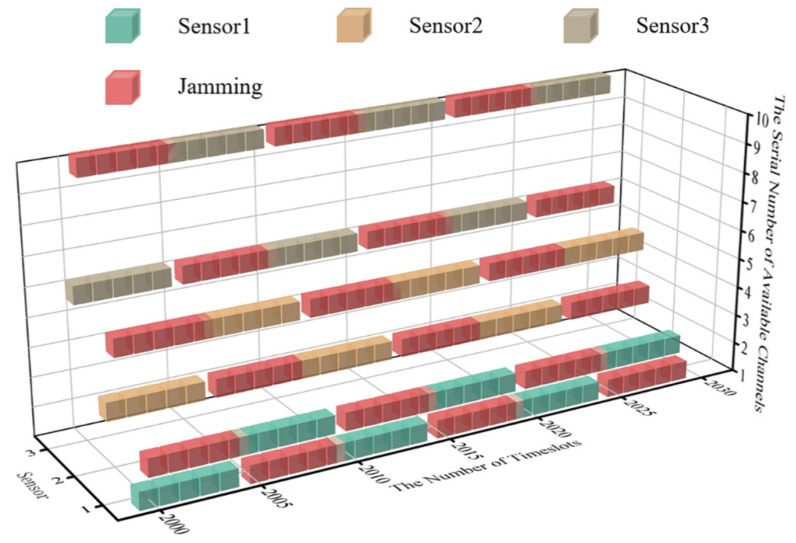
The cubic diagram of DAJA’s convergent state with time delay.

**Table 1 sensors-22-08159-t001:** Settings of model-related parameters.

Parameters	Value
Communication nodes N	3
The number of jamming’s selection J	3
Adjacent nodes M	10
Timeslot $T_{s}$	0.3 ms
Sub-slot $T_{observe}$	0.1 ms
Sub-slot $T_{act}$	0.2 ms
The number of slots jamming lasts L	5
Total timeslots $N_{S}$	10,000
Transmission reward E	1
Learning factor α	0.8
Discount factor β	0.6
Discount factor γ	0.1

**Table 2 sensors-22-08159-t002:** ARTEC in different timeslot periods among mentioned algorithms.

Periods	Cycles	Algorithm		
		DAJA	JMAA	OFH
403~407	403	14	5	9
	404	13	8	9
	405	15	14	9
	406	15	9	9
	407	15	10	9
455~459	455	15	12	9
	456	15	10	9
	457	15	15	9
	458	15	15	9
	459	15	15	9

**Table 3 sensors-22-08159-t003:** ARTEC in different timeslot periods between DAJA and JMAA [12].

	Numerator	DAJA	JMAA
Denominator			
DAJA		1	0.89
JMAA		1.13	1

## Data Availability

Not applicable.

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
