# Peer review of "A Distributed Anti-Jamming Algorithm Based on Actor–Critic Countering Intelligent Malicious Jamming for WSN"

_sensors, 2022, doi:10.3390/s22218159_

Round 1

Reviewer 1 Report (New Reviewer)

Please find the attached report.

Reviewer Recommendation Term:     Major revisions

Author Response

Reviewer 2 Report (New Reviewer)

Please see attached my recommendation and suggestion

Author Response

Reviewer 3 Report (New Reviewer)

The authors have conducted a worthwhile piece of work. The optimization part is missing. 

Round 2

Reviewer 1 Report (New Reviewer)

Thanks for the resubmission. The authors have addressed all comments and suggestions. I recommend accepting. All my concerns were addressed. But there are some minor comments.

(i)- Replace the word " (assume)" with propose. This word is frequently repeated in the revised version article. Try to replace it with another word to give the best quality.

(ii)- There are many abbreviations. Please reduce the abbreviations. There are many abbreviations in the modified paper. I suggest that you reduce some of the abbreviations to give readers and other researchers ease and understanding of the article.

(iii)- Provide the abbreviation "(SARSA)" State–action–reward–state–action (SARSA) algorithm in the revised manuscript.

Reviewer Recommendation: Accept in present form

Reviewer 3 Report (New Reviewer)

The authors have responded to individual queries asked by the reviewer. I recommend possible publication in this esteemed journal.

This manuscript is a resubmission of an earlier submission. The following is a list of the peer review reports and author responses from that submission.

Round 1

Reviewer 1 Report

(1)There is generally no need to write acronyms in the title.

(2)Line 90 describes that a timeslot is divided into three parts, but the subsequent explanation and Figure 2 only explain two parts.

(3)The coordinate axis in Figure 9 is described in Chinese.

(4)The authors are recommended to clearly show the advantages of the proposed scheme by comparing it with the traditional non machine learning scheme.

(5)In the system model, if the user uses the same channel with other users or interference, it is considered that the transmission fails. Otherwise, it is considered that the transmission is successful. Is the modeling too simple? In such a simple modeling, is it necessary to use reinforcement learning? From the convergence results in the simulation, it seems that the channel without interference is selected according to the observation results.

(6)All the figures in the paper are very vague, which seems to be in the wrong format.

Reviewer 2 Report

The submitted work corresponds to the introduction of a new anti-jamming algorithm. Please consider the following remarks:

1) Although the overall technical contribution is significant, the English of the paper are extremely poor.

2) At line 34 please change "Related works" to "Related work".

3) At line 357 please change "Further thingking" to "Further analysis and future work".

4) At line 225 please remove the word "the".

5) At section 4.1, Table 1, how the parameters values of the simulation model have been selected? What are the criteria for selecting these values?

6) The simulation results presented at section 4 are succesfully compared with published literature.

7) Abstract and conclusions sections are fine.

8) A few more references should be used.

Reviewer 3 Report

The paper models the multi-agent transmission system as a Multi-Agent Markov Decision Process and proposes a Distributed Anti-Jamming Algorithm based on Actor-Critic.

It is difficult to follow the paper and it needs a proof read.

Too many typos and grammatical errors such 

“This paper is contribution is as follows:”

The algorithm is not written well. What is the output of it?

Which simulation environment did you use?

Reviewer 4 Report

1. The writing style is bad -- should have a professional technical editor to significantly polish this paper.

2. There is no validation of your approach. You should validate your simulation by either analytic modeling or measurements.

3. Can anyone actually implement your algorithm in real hardware? You should have a prototype.

4. The simulation is over-simplified. What happen if N>3? You just use standard machine learning algorithm with trivial parameter selection. Everyone is using machine learning. What's your contribution on the AI part?

5. You did not provide network throughput comparison as in the following paper

A multi-agent reinforcement learning anti-jamming method with partially overlapping channels

  Yunpeng Zhang,Luliang Jia,Nan Qi,Yifan Xu,Xueqiang Chen  

6. could you provide your simulation code in a supplementary document to convince the reviewers?

Round 2

Reviewer 1 Report

The author corrected some typos, and I have the following comments.

(1) The author does not directly give the description and comparison of traditional schemes.

(2) The author does not explain the simulation results, that is, the user only needs to select the channel without interference by observation.Such a simple result makes this problem seem unnecessary to use reinforcement learning to solve.

Reviewer 3 Report

The revised manuscript is better.

Reviewer 4 Report

This paper is still weak. The authors should  keep their promise to implement the proposed scheme. Otherwise, this paper is useless.